# Comparative Study on Volunteering among Older Korean Immigrants in the United States and Older Koreans in South Korea

**DOI:** 10.3390/ijerph19127297

**Published:** 2022-06-14

**Authors:** Miya Chang

**Affiliations:** The Center for Multicultural Studies, Daegu Catholic University, Gyeongsan 38430, Korea; philsaram@cu.ac.kr

**Keywords:** volunteering, older Korean immigrants, older Koreans, social participation, aging, cultural factors

## Abstract

This study aimed to determine the most important factors that influence the prevalence of volunteering among older Koreans in the United States and in Korea and to identify sociodemographic resources, cultural resources, and social resources that are correlated with volunteering. The data were gathered from older Koreans aged 60 or over from the two countries (*n* = 480). The prevalence of volunteering was 23.3% for older Korean immigrants in the United States versus 14.7% for the older Koreans in Korea. This study found that there was a significant relationship between human capital (individual resources) and volunteering in both countries. Education and financial status had a positive relationship with volunteering among older Koreans in both countries. Cultural resources and social resources were the only important predictors of volunteering among older Korean immigrants in the United States. Regarding types of volunteering, older Koreans in both countries were more likely to participate in informal volunteering than formal volunteering. These findings differ somewhat from those reported by previous studies. This study was the first attempt to examine older adults from these two countries with a focus on the correlation between cultural factors, social resources, and volunteering.

## 1. Introduction

Volunteering has marked benefits for older persons who do volunteer work. It has a positive effect on well-being and plays an important role in a successful and productive navigation of advanced age [1,2]. Volunteering is usually defined as unpaid work that benefits other individuals (e.g., giving money and goods, providing unpaid assistance, and offering emotional support to other people). Volunteering is also considered beneficial for older adults because it can provide a means of acquiring substitutes for work and family roles [3]. In light of increasing life expectancy, volunteering among older adults is increasingly a salient issue and popular activity. Therefore, volunteering is one important form of productive civic and social participation for older adults that has consistently influenced social work practice and gerontology.

In many countries, the proportion of the population is increasingly filled with retired adults who now contribute to the economy through unpaid work [4]. According to the Corporation for National Community and Service (2016), volunteering among American adults aged 55 and older contributed more than 77 billion USD to their communities in 2015. As a consequence, volunteering contributes to world economic status [5].

Previous studies have reported that volunteering provides social support, a sense of belonging, and access to information that can be crucial for mental health [6,7]. Researchers have also investigated the relationships between volunteering and the well-being of older adults, finding that volunteering as one type of social engagement offers numerous benefits [8]. Furthermore, several studies have suggested that volunteering is correlated with better mental health, lower morbidity, lower utilization of health services, reduced mortality in later life, and the maintenance of previous activity patterns over time among active volunteers compared to non-volunteers [9,10]. Moreover, volunteers tend to be healthier and more socially integrated individuals than non-volunteers [8].

Resource theory is a suitable theoretical framework for examining volunteering activities in older adults. It has been widely recognized that having resources is one of the main predictors of participation in volunteering. The resource theory framework posits that volunteering requires individuals to have resources that enhance volunteering opportunities [11]. Specifically, socioeconomic resources, such as education and income, are strongly related with volunteering [12,13].

Most studies on volunteering are conducted in the United States or selected Western European countries [14,15]. However, few studies have explored the unique volunteering experiences of culturally and linguistically diverse older adults. Furthermore, volunteering studies have focused on the general population rather than on specific ethnic and immigrant groups in the United States. A few studies on volunteering among ethnic minorities have noted that many ethnic communities have a tradition of mutual help among close family and friends and thus are more likely to participate in informal volunteering outside of well-recognized formal volunteer organizations [16]. In addition, formal volunteering is defined as unpaid work for formal organizations and agencies [16].

Relatively few studies on volunteering among older Asian immigrants exist and little is known about what factors are correlated with hindering or facilitating their volunteering [17]. Despite a rapid increase in the number of older Koreans living in both the United States and South Korea (hereafter ‘Korea’), there is no comparison study focusing on the rate of volunteering in this population. This paper is the first attempt to examine older adults from these two countries with a focus on the correlation between cultural factors, social resources, and volunteering.

To address the main gaps in the existing literature, the purpose of this study is to explore the differences and similarities in volunteering among older Korean immigrants in the United States and older Koreans in Korea. Thus, this study aims to determine the most important factors that influence the prevalence of volunteering among older Koreans in both countries and to identify sociodemographic characteristics, cultural factors, and social resources that are correlated with volunteering.

## 2. Literature Review

### 2.1. Prevalence of Volunteering

According to the US Bureau of Labor Statistics (2016), the three sectors where volunteers worked most often were religious sectors (33.1%), educational organizations (25.2%), and social community service organizations (14.6%). In this set of statistics, about one-fourth of older adults (23.5%) aged 65 years and over participated in volunteering. In addition, men and women participated in different main activities when they volunteered in 2015. For example, men were most likely to engage in general labor, while women were most likely to collect food or do fundraising.

Meanwhile, volunteering among older Asian immigrants is lower compared to nonimmigrant older adults in the United States [18]. For example, 26.5% of older adults in the United States participated in volunteering in 2012 [19]. On the other hand, it is estimated that 19% of older Asian immigrants aged 65 years and older participated in volunteering in 2009 [20].

Previous studies indicate that the rate of volunteer work among older Asian immigrants is still lower than Whites in the United States due to their difficulties with adjusting to life in the United States. For example, the US Census Bureau Survey (2011) showed that 19.6% of Asian Americans volunteered in 2011, compared to 27.8% of Whites [21]. A recent study reported that 24% of older adults in the United States participated in volunteering [1].

In Korea, the 2019 Time Use Survey recently found that 6.5% of older Koreans participated in volunteering. The results showed that both male and female older adults who are healthy tend to increase their volunteer hours. Another study found that approximately 4% of older Koreans in Korea participated in volunteering using the data of the 2017 National Survey of Senior Citizens [22]. As a result, the overall rate of volunteering was estimated to range from 2% to 7% in Korea. Therefore, the volunteering rate among older Koreans in Korea was generally lower than their participation rate in the United States.

### 2.2. Sociodemographic Resources and Volunteering

Relatively few studies on volunteering among older Koreans exist. Several studies found a positive correlation between individual sociodemographic resources, such as education, income, and health status, and volunteering [16,23]. Numerous studies have demonstrated that educational attainment is positively correlated with volunteering among older adults [13,24,25]. For example, Lee et al. (2018), using data from a probability survey, reported older Asian immigrants with a higher level of education and those who became United States citizens were more likely to engage in volunteering [25]. Therefore, higher levels of education are associated with a greater likelihood of volunteering.

Household income and health status are additional potent predictors of volunteering among older adults. Household income is an asset that makes it possible for people to be interested in volunteering [26,27,28,29]. Although available empirical results are ambiguous, a positive correlation between income and volunteering has been found in several studies. In addition, health status consistently predicts volunteering within the general older population and older immigrant population. Older adults in good health are more likely to engage in volunteering than their counterparts with poor health conditions as in most other countries [23,30,31].

In Korea, a few studies have found a positive correlation between individual sociodemographic factors and volunteering [32,33]. As a result, older adults with higher socioeconomic status, including higher educational levels and healthier health status, are more likely to volunteer because they are more likely to be asked to volunteer. For example, healthy older adults were more likely to volunteer than unhealthy people, and college graduates or higher were more likely to volunteer than middle school graduates or those of lower education [33]. Participation in volunteer activity differed significantly depending on age, educational level, economic status, health status, ability use information, social networks, and frequency of contact [22]. Specifically, social networking and contact frequency were shown to be significant effect factors.

### 2.3. Cultural Resources and Volunteering

Culture plays a significant role in social behavior. Differences in culture tend to influence the way people live, the choices they make, what they think, and whom they associate with. The culture of a society refers to the extent to which a group or an individual adopts the values and practices of their cultural norm [34]. Cultural factors such as values and attitudes are important factors in how individuals perceive and manage life events. Culture and religion are inseparable and always exist in a close relation [35]. Religion influences culture and it also constitutes culture [36]. Religion as a cultural identity marker is a cultural tradition.

For many Koreans, cultural values such as filial piety, elder care, the preservation of face, and social sanctions for enforcing moral standards influence their decisions about life events [37,38]. According to traditional Asian cultural values, caring for older adults is a responsibility that fulfills filial piety [39]. Another cultural difference concerns collectivist values. In many Asian cultures, elder care, gift-giving, and charity are rooted in culturally collectivist rather than individualist values. This therefore influences how many Asians view volunteering and participation in volunteer work.

Religion and volunteering are strongly connected. The dimension of religiosity is significant in predicting volunteering [40,41,42]. According to a review of Gallup data from 145 countries (15 Western European countries), those who had attended a religious worship service in the past week were more likely to engage in volunteering. With regard to religious affiliation, older adults who identified as religious were more likely to volunteer than those who did not so identify [43,44]. Certainly, religion cultivates ideas of generosity and care toward others, and serves as a resource to support volunteering. Therefore, religious persons consistently showed higher rates of volunteering than those who did not have any religion. Volunteering for religious and secular organizations by ethnic and immigrant groups is essential as a cultural factor.

### 2.4. Social Resources and Volunteering

Social resources refer to social support and social networks, social activities, and the availability of social contacts that enable people to have opportunities to participate in volunteering. Social networks provide information and enable people to have opportunities for volunteering. Few studies have focused on the importance of social support and social networks in promoting volunteering benefits among older adults [45,46]. Older adults who have maintained social networks are correlated with an increased likelihood of volunteering [46].

Previous studies reported that Asian immigrants’ social networks have great influence on their participation in volunteering [47] because social networks facilitate effective organizations and build up a level of trust sufficient to persuade individuals to volunteer. A greater availability of social networks is positively correlated with volunteering in older adults [31]. In this line of study, the effect of acculturation is one of the most determinant factors in volunteering work for ethnic minorities [25]. Acculturation is a process of critical cultural adjustment to new environments in which immigrants acquire the behaviors, languages, social values, norms, beliefs, attitudes, and social affiliations of their new country.

In Korea, the main support for older adults has not been welfare coverage of the Korean government but family or close kin [48]. Therefore, older adults may prefer to contribute their time and energy to the family, as they are rooted in culturally expected collectivist attitudes. Applying social resources, such as group members or family ties, to older Koreans in Korea may not seem entirely appropriate. For example, strong family ties may limit productive activities of older adults outside the family. However, compared to Western countries, few studies have investigated factors related to volunteering among older Koreans in Korea. Therefore, it is unknown if these finding would be valid in countries in Asia such as Korea.

### 2.5. Theoretical Foundation and Hypotheses

Resource theory is used to explain volunteering activity. The theory assumes that volunteering requires individuals to have resources that facilitate involvement in volunteering and enhance volunteering opportunities [11]. Resource theory includes individual capital (such as education, income, health, and marital status), cultural capital (such as religious affiliation and attitude toward elder care), and social capital (such as social support and social networks). Previous studies have found that these resources are strongly correlated with volunteering activities [12,49].

With resource theory as its conceptual framework, this study seeks to answer three main questions: (1) What is the prevalence of volunteering of older Korean immigrants in the United States and older Koreans in Korea? (2) What are the most important factors that influence the volunteering experience among older Koreans in both countries? (3) Are relationships between socio-demographic resources and volunteering moderated by social resource variables (e.g., social support and social networks)?

Thus, this study proposed the following hypotheses: (1) Those with higher educational attainment will show higher rates of volunteering than those with lower educational attainment. (2) Those who have religious affiliation will show higher rates of volunteering than those who have no religious affiliation. (3) Those with greater social resources (social support and social networks) will show higher rates of volunteering than those with lower social support and social networks.

## 3. Methods

### 3.1. Participants

The 480 study participants were older Koreans aged 60 years or older and residing either in Los Angeles County or in Korea in 2017. The study’s participants were recruited based on the following criteria: (1) self-identification as a Korean and Korean immigrant; (2) age ranged from 60 years to 79 years at the time of data collection (workers in Korea tend to retire early, prior to the age of 60 years. The formal retirement age in Korea is 60); (3) place of birth: Korea; and (4) location of current residence: Los Angeles in the United States; Seoul or Daegu in Korea (two metropolitan cities).

### 3.2. Instruments

#### 3.2.1. Volunteering

Volunteering was measured by dichotomous Yes (1) or No (0) answer choices. Respondents were asked: “In the past 12 months, have you done any volunteer work or community service that you have not been paid for?” Regarding types of volunteering, formal volunteering was measured by the question: “In the past 12 months, have you served as a volunteer on any local board, council, or organization?” (Answer: Yes/No) and informal volunteering was measured by the question: “In the past 12 months, have you served as a volunteer providing help to others such as friends, neighbors, or relatives (e.g., help with practical household thing, looking for children, giving advice or talking with someone who needs cheering up)?” (Answer: Yes/No).

#### 3.2.2. Sociodemographic Characteristics

As personal resources, education was reported in five categories: below high school, high school, college, graduate, or other. Income was reported in six categories, from below 1000 USD per month to 5000 USD or more per month (Korean Won converted to USD). For financial status, a five point Likert-type scale ranging from “very bad” to “very good” was answered, with higher scores indicating better financial status. For measurement of health status, a five point Likert-type scale ranging from “very poor” to “excellent” was answered, with higher scores indicating better health status.

#### 3.2.3. Cultural Resources

Attitude toward elder care was measured using 10 statements [50]. The respondents answered 10 questions about the following: “Adult children should live together with their elderly parents at the parents’ wish” “It is wrong to place an impaired parent in a nursing home”, “Elderly parents should make the final decisions on important family matters”, “Adult children should obey the decisions of their parents even when they think the decisions are bad”, “A daughter-in-law should have more responsibilities than married sons to care for the elderly parents”, “Adult children concerned about the well-being of their elderly parents should keep bad family news from their parents”, “Adult children and their spouses who live with elderly parents should take of all household chores to ensure that their parents are free from housework”, and “Adult children who live far away from their elderly parents should call or write to their parents regularly, at least once a week”. Using a five point Likert-type scale, the question was answered ranging from (1) “strongly disagree” to (5) “strongly agree”, with higher scores representing a more positive attitude toward elder care. The Cronbach’s alpha for elder care was 0.89 for the present study’s sample.

#### 3.2.4. Social Resources

Social networks were measured using a six-item questionnaire developed by Lubben Scale-6 [51]. The respondents answered six questions about “How many” family member and friends “do you see or hear from at least once a month”, “do you feel at ease with that you can talk about private matters”, “do you feel close to such that you could call on them for help”, “do you see or hear from at least once a month”, “do you feel at ease with that you can talk about private matters”, and “do you feel close to such that you could call on them for help”. The question was answered ranging from (1) “no one” to (6) “nine or more”, with higher scores representing greater social networks. The reliability of the social networks scale was 0.83, as measured by Cronbach’s alpha.

Perceived level of social support was measured using 10 items in the survey. The respondents answered 10 questions about “How often” during the past 30 days someone was available “to give you good advice about a crisis”, “to take you to the doctor if you needed it”, “to have a good time with”, “to confide in or talk about yourself or your problems”, “who shows you love and affection”, “to prepare your meals if you were unable to do it yourself”, “to help you with daily chores if you were sick”, “to turn to for suggestions about how to deal with a personal problem”, and “to love and make you feel wanted”. Using a five point Likert-type scale, the question was answered ranging from (1) ‘seldom’ to (5) ‘always’ with higher scores representing greater social support. The Cronbach’s alpha for the social support scale was 0.82 for the present study’s sample.

### 3.3. Data Collection

A convenience sampling method was employed to recruit study subjects. To reduce convenience sampling bias inherent in non-probability, a number of sampling sites were used to recruit study participants in both countries. In addition, quota sampling divides a population into various categories, such as demographic characteristics (e.g., age, gender) and involves setting quotas on the number of elements to be selected from each category. Therefore, quota sampling is an improvement over convenient sampling because the researcher can ensure the inclusion of diverse elements of the population. To enhance the diversity and representativeness of this study, participants are categorized by age (60–69 and 70–79) and gender (male and female) using the quota sampling method. The researcher provided informed consent and verbally explained the study to the potential participants. Informed consent was obtained from all participants involved in the study. As a result, the researcher was able to collect by the individual approach (380 participants) and the group approach, such as local organizations and institutions (100 participants) in both countries.

Data were collected by using a structured questionnaire in Korean. Respondents took about one hour to complete the questionnaire. To minimize the refuse rate, respondents were offered five USD cash incentive for their participation in this study. The acceptance rate was 92% for the sample. This study includes all data from participants aged 60 to 79 years, excluding 42 subjects who refused to answer and missing cases. Korea is one of the most rapidly aging countries. However, workers tend to retire early, prior to the age of 60 years. The survey questionnaire was in Korean for older adults and it was developed through back translation. The first translator translated the survey questionnaire into Korean and then the second translator translated Korean into English.

### 3.4. Ethical Consideration

The researcher first addressed the purpose of the study, the procedures to be used, the duration of the survey, the right to refuse to answer any question, and the researcher’s contact information. All participants voluntarily signed the consent form after the purpose of the study were explained. Before beginning the data collection, IRB approval for the present study was obtained from the University of California, Los Angeles (UCLA #16-001931).

### 3.5. Data Analysis

Frequency distributions were used to summarize the sample characteristics, perceived level of social support and social networks, attitude toward elder care, and participation in volunteering of the study participants. Multicollinearity and bivariate correlations were checked. Bivariate analyses were conducted to examine the relationship between volunteering and the independent variables. This study includes those who answered all questions. All analysis was conducted in Stata 14.

## 4. Results

### 4.1. Descriptive Analysis

Descriptive statistics of the sample are displayed in Table 1. For older Korean immigrants in the United States, the mean age was 69.4 (*SD* = 5.2) with a 60–79 range, the mean score of social support was 20.6 (*SD* = 6.5), of social networks 13.7 (*SD* = 4.8), and of attitude toward elder care, 28.8 (*SD* = 3.9). For older Koreans in Korea, the mean age was 68.9 (*SD* = 5.7) with a 60–79 range, the mean score of social support was 26.7 (*SD* =10.3), social networks, 17.3 (*SD* =6.6), and attitude toward elder care, 24.8 (*SD* = 4.9). Regarding income, religion, health status, and perceived financial status, there were similar patterns in both countries. However, education and income sources were significant differences between the two countries. Regarding education levels, approximately 39% of older Korean immigrants in the United States had “above high school” education, while 25% of their counterparts in Korea had “above high school” education. As for income sources, about one-third of older Korean immigrants (60.0%) in the United States were dependent on public assistance (social welfare), and 3.3% received assistance from adult children, while slightly more than one-third of older Koreans in Korea were dependent on public assistance and one-fourth (26.7%) received assistance from adult children. Regarding the prevalence of volunteering, about one-fourth of older Korean immigrants (23.3%) in the United States did volunteer work versus 14.2% of older Koreans in Korea.

### 4.2. Bivariate Analysis 

As shown in Table 2, chi-square analysis was completed. Education, income, marital status, health status, and perceived financial status were all statistically significantly correlated with volunteering among older Koreans in both countries. Religion, attitude toward elder care, social support, and social networks were statistically significantly correlated with volunteering only for older Korean immigrants in the United States, while age was statistically significantly correlated with volunteering only for older Koreans in Korea. 

#### 4.2.1. Volunteering Activities in the United States

Regarding education levels, 35.6% of older Korean immigrants in the United States with “college” education were volunteers, while 11.3% of older Korean immigrants with “high school” education were volunteers. Regarding income, 40.7% of older Korean immigrants in the United States with between 2000–2999 USD monthly income were volunteers, while 21.8% of older Korean immigrants with monthly income between 1000–1999 USD monthly income were volunteers. Regarding marital status, 29.3% of older Korean immigrants with “married” were volunteers versus 7.3% of older Korean immigrants identifying as “separated”, “divorced”, “widowed”, or “not married” were volunteers. Regarding health status, about half (47.9%) of older Korean immigrants in the United States with “good” or “very good” health status were volunteers, while 9.2% of older Korean immigrants with “poor” health status were volunteers. With regard to perceived financial status, 65% of older Korean immigrants in the United States with “good” or “very good” perceived financial status were volunteers, while 6.3% of older Korean immigrants with “bad” perceived financial status were volunteers. With regard to religion, 26.6% of older Korean immigrants in the United States who identified as religious were volunteers. Regarding social networks, 85.7% of older Korean immigrants with “3 or more” contacting among family members and friends were volunteers, while 29.8% of older Korean immigrants with “1 or 2” contacting among family members and friends were volunteers. Regarding social support, 51.2% of Korean immigrants who “often” had someone available to help them were volunteers versus 16.8% of Korean immigrants who “seldom” had someone available. Regarding attitude toward elder care, 14.3% of older Korean immigrants who selected “agree” with statements such as respecting older adult’s decision and caregiving were volunteers versus 31.3% of older Korean immigrants who selected “disagree” with such statements. In addition, those who received public assistances (i.e., social welfare recipients) were more likely than employed older adults to have participated in volunteering activities (26.5% and 18.3%, respectively). Regarding living arrangements, those “living with spouse only” and those “living with grandchildren only (under 18 years)” were more likely than those “living alone” to have participated in volunteering (30.2%, 11.1%, and 3.8%, respectively). 

#### 4.2.2. Volunteering Activities in Korea

Regarding education levels, 24.5% of older Koreans in Korea with “college” education were volunteers, while 11.4% of older Korean Koreans with “high school” education were volunteers. Regarding income, 25.6% of older Koreans in Korea with monthly income between 2000–2999 USD monthly income were volunteers (Korean Won converted to USD), while 11.5% of older Koreans with monthly income between 1000–1999 USD monthly income were volunteers. Regarding marital status, 18.5% of older Koreans identifying as “married” were volunteers versus 5.1% of older Koreans identifying as “separated”, “divorced”, “widowed” or “not married. Regarding health status, 33.3% of older Koreans in Korea with “good” or “very good” health status were volunteers, while 4.4% of older Koreans with “poor” health status were volunteers. With regard to perceived financial status, 26.3% of older Koreans in Korea with “good” or “very good” perceived financial status were volunteers, while 1.9% of older Koreans with “bad” perceived financial status were volunteers. In addition, those who received public assistance (i.e., social welfare recipients) were more likely than employed older adults to have participated in volunteering activities (21.7% and 6.9%, respectively). Regarding living arrangements, those “living with spouse only” and those “living with grandchildren only (under 18 years)” were more likely than those “living alone” to have participated in volunteering (17.5%, 20.2%, and 9.1%, respectively).

#### 4.2.3. Types of Volunteering and Volunteering Hours

Regarding formal volunteering, 5.4% of older Korean immigrant in the United States reported formal volunteering, while 9.6% of older Koreans in Korea reported formal volunteering (Table 3). Regarding informal volunteering, older Korean immigrants in the United States were twice as likely to report informal volunteering as older Koreans in Korea (25.8% and 13.8%, respectively). Regarding volunteering hours, this was assessed using the number of hours volunteered per month. Older Korean immigrants in the United States reported volunteering more than twice as many hours as older Koreans in Korea.

## 5. Discussion

There is a rich body of literature on the predictors and outcomes of volunteering among the general population of older adults in the United States. However, few studies have explored the unique volunteering experiences of culturally and linguistically diverse older adults. This is the first study examining the similarities and differences in volunteering among older Korean immigrants in the United States and older Koreans in Korea. In the light of limited comparative studies on volunteering among older Koreans, this study explored various determinants and the prevalence of volunteering of older Koreans in the United States and in Korea.

Specifically, one of the most interesting findings is that the prevalence of volunteering among older Koreans in both countries is significantly different. Regarding the prevalence of volunteering, 23.3% of older Korean immigrants in the United States did volunteer work versus 14.2% of older Koreans in Korea. Interestingly, the prevalence of volunteering among older Korean immigrants in the United States is similar to that among the older general population in the United States. These findings are consistent with prior evidence which noted that older adults in Korea are less likely to participate in volunteering than older Korean immigrants in the United States [22]. One possible explanation for this result may be that the primary means of support for older adults in Korea has not been the welfare provision of the Korean government but the family. Older adults may therefore prefer to contribute their time and energy to the family as they are accustomed to strong family ties.

Regarding types of volunteering, older Koreans in both countries are more likely to participate in informal volunteering than formal volunteering. One possible explanation for this result may be that the higher frequency of informal volunteering is due to acculturation, such as language availability and lack of information about formal organizations. Another possible explanation for this result may be that the higher frequency of informal volunteering is due to barriers engaging in formal volunteering that need be considered the centrality of the problems of transportation and personal resources. Regarding language proficiency among older Korean immigrants, opportunities for participation, in reality, are not equal for other groups. In addition, language barriers may exclude ethnic older adults from participation in formal volunteering opportunities. Ethnic minorities may also experience limited access to volunteering infrastructures in mainstream society.

Not surprisingly, volunteering is statistically significantly correlated with individual resources, such as education, income, marital status, health status, and perceived financial status among older Korean immigrants in the United States and older Koreans in Korea. Surprisingly, according to the results of this study, religion, living arrangements (living alone, living with spouse only), attitude toward elder care, social support, and social networks are the only important predictors of volunteering among older Korean immigrants in the United States while age and income resources are the only important predictors of volunteering among older Koreans in in this study. More research is needed to further examine the reason for these differences. These findings are not consistent with prior evidence that noted older adults who had attended a religious worship service in South Korea was more likely to participate in volunteering than those who did not have any religion [42].

Regarding income resources, social welfare recipients were more likely to participate in volunteering than employed older adults in return for their benefits. However, while prevailing values among the general public are well known, the views of social welfare recipients on volunteering have rarely been considered in assessing participation in volunteering. Thus, there is a conventional assumption that those who receive social welfare benefits are somewhat less supportive of reciprocity than the general public as a whole. This assumption may be incorrect; however, more research is needed to further examine the reason for these differences.

This study found that there is a significant relationship between financial status and volunteering. Financial status was the most salient predictor of volunteering among older Koreans in both countries. Finally, those with higher financial status showed higher rates of volunteering than those with lower financial status. This finding is consistent with previous studies that suggest persons in a more stable financial position seem to be more likely to engage in volunteer work [28]. One possible reason for this is that persons with greater financial status can make more of a commitment of time and resources to volunteer work. Having a higher household income can also provide a sense of stability and security that allows people to spend time on volunteer work. However, Korea has instituted a mandatory retirement policy. Unlike a few Western countries with strong social security systems, many low-income older adults in Korea find themselves in an economically vulnerable situation after retirement.

Consistent with previous studies, this study also found a positive correlation between education and volunteering. Those with higher educational attainment showed higher rates of volunteering than those with lower rates. This finding supports the first hypothesis of this study. Volunteers did have higher levels of education compared to the non-volunteers [26]. That levels of educational attainment should be correlated with volunteering is an important empirical finding. Previous studies have shown that educational attainment may be the most significant predictor of volunteering in contemporary society because educated people are more likely to join volunteer organizations [13,26]. Educational attainment also influences participation in volunteer work among older Koreans in both the United States and Korea. It may be that education plays a role in increasing opportunities for older adults to volunteer. Older adults with more education may have higher expectations, obligations, or contributions to make to society. This result is in keeping with the findings of earlier studies, as noted above, as well as the first hypothesis of this study. The effect of education and financial status on older Koreans is exactly as predicted by the conventional resource theory of volunteering. These individual resources turn out to be resources for older Koreans in both countries.

Another interesting finding is that there was no statistically significant relationship between religious affiliations and volunteering in Korea. According to previous studies, older adults are more likely to be involved in religious organizations and their community and less likely to be involved in educational and public programs. However, this study did not find that religious individuals volunteered more than nonreligious individuals. However, there was statistically significant relationship between religious affiliations and volunteering in the United Sates. This finding partially supports the second hypothesis of this study. Previous studies found that Korean immigrants in the United States were more likely to volunteer for religious institutions that are ethnically homogeneous [16]. Religious affiliation has a more powerful effect on older Korean immigrants’ volunteering than older Koreans’ volunteering. Religious activity provides the cultural resource on which volunteering depends. Thus, religion does help explain some of the difference in volunteering between older Korean immigrants and older Koreans.

An additional finding in this study is that social resource variables and volunteering tend not to be significantly correlated in Korea. This finding is not consistent with previous studies that suggested that those who have greater social resources show a higher rate of volunteering than those with lower social resources. One possible explanation for this result may be that older Koreans in Korea have a distinct social context, environment, or other resources which may influence the results. According to a previous study, social resources act as assets that influence the act of giving services to others [29]. The results in this study point out the under-researched importance of taking into account the recruitment potential of older adults. This finding partial support the third hypothesis in this study.

In conclusion, a major contribution of this study is that it represents one of the first comparative studies on volunteering among older Koreans in the United States and in Korea. Therefore, comparing volunteering rates among older Koreans in the United States and Korea was meaningful in this study. Interestingly, older Korean immigrants in the United States were more likely to participate in volunteering than those in Korea.

Some limitations of the study need to be considered. First, the sample was obtained using convenience sampling. The findings of a convenience sampling set limitations on the generalizability of the data. Therefore, caution should be used in generalizing the findings in this study to the full range of the aging Korean population in both countries. Second, primary data cross-sectional design only allowed testing of correlations, and therefore has a very limited ability to establish a causal relationship between variables. In the future, longitudinal data should be explored to investigate the causal relationships between individual characteristics and volunteering activity among older Koreans in both countries. Third, this study has a limited ability to assess some variables (e.g., types of volunteering and hours of volunteering) in the primary survey data and only examined urban area samples. Due to data limitations, this study is unable to identify how often someone volunteered in the past 12 months. Therefore, the limitations of sample size and the availability of data relevant to volunteering somewhat constrain the findings.

## 6. Conclusions

Given the growing number of diverse older adults in the United States and the importance of optimizing their contributions to society, there is a need for studies such as one, which investigated the similarities and differences of volunteering among older Koreans in the United States and in Korea. This study calls attention to comparative studies of volunteering among older Korean immigrants in the United States and older Koreans in Korea. This first comparative study of volunteering among older Koreans in both countries emphasized explanations for similarities and differences. Clearly, this population deserves greater attention in aging research, policy and practice. Additionally, systematically tracking immigration-related characteristics among a growing Asian immigrant population in the United States will enable researchers to better understand the influence of socio-demographic characteristics on volunteering among older Korean immigrants in the United States.

The study results address implications for practice and policy in the United States and in Korea for those working with older adults. The bivariate results suggest that the rates of prevalence of volunteering among older Korean immigrants in the United States are not comparable with those in Korea. These findings indicate that practitioners should be aware of the considerable prevalence of volunteering among older Koreas immigrants in the United States. For example, 23.3% of older Korean immigrants in the United State did volunteering versus 14.2% of older Koreans in Korea.

This study suggests that financial status is the most powerful predictor of volunteering. This finding implies that practitioners should design volunteer opportunities among low-income older adults, such as stipended volunteer roles. Specifically, a stipend is some level of financial remuneration paid to an individual for participating in volunteer work. This monetary compensation is not designed to be equivalent to labor market income and should still meet the basic criteria for volunteerism. For example, since the Edward Kennedy Service America Act in the United States was signed into law in 2009, stipended volunteering roles in national service programs, such as AmeriCorps and the Peace Corps, dramatically increased. Volunteers in stipended roles across the United States receive monetary support that is considerably below market wages. Similarly, stipended young adult volunteers are more likely to finish the academic year than non-stipended volunteers, which suggests that stipends may help increase volunteer retention.

Another study found that older Korean immigrants in the United States tend to have higher levels of education than those in Korea. Indeed, previous studies have proven that educational attainment has an impact on volunteering in one’s later life. Not surprisingly, high levels of education boost volunteering rates among older Korean immigrants in the United States because of the greater proportion of highly educated older adults. This implies that there is a larger number of potential volunteers, because 38.4% of older Korean immigrants in the United States have at least college graduate-level education compared to 28% of those in Korea. Therefore, practitioners should endeavor to assess how older adults might be influenced to initiate volunteering among older Koreans in both countries. Exploring older Koreans’ volunteering patterns would help practitioners understand the relationships between individual and social characteristics, and volunteering. In addition, practitioners can provide possible volunteer opportunities and could increase engagement in volunteer work for older Koreans in both countries through a collaborative coordinated system, developing programs, and launching media and public awareness campaigns as an institutional support. Most volunteers are recruited in person and by the media.

Furthermore, practitioners should design a wide range of more tailored opportunities to develop volunteer programs that can improve the psychological well-being of volunteers in later life, increase prevalence rate, enhance the recruitment method of potential volunteers, and strengthen communities. In addition, practitioners could develop policies to encourage faith-based volunteering and more committed volunteer participation through culturally appropriate and competent promotional programs with religious affiliations to increase older Koreans’ engagement in volunteer work in their own ethnic communities and beyond.

After recruitment, important strategies are needed for retaining volunteers, and the study results will aid in developing those strategies, such as accessible and affordable transportation, health care, safety, and community cohesion opportunities. To recruit and motivate volunteers, organizations need to understand the different contexts in both countries. Above all, in order to retain current volunteers, diverse recreational, innovative, and health promotion programs should be developed to continue volunteering for older adults who are satisfied with volunteering and feel a benefit from volunteering.

Specifically, attention must be paid to the barriers to volunteering in Asian communities as well as to Asian cultural differences that play a role in how individuals participate in volunteering among older Korean immigrants in the United States. It is important to develop more diverse volunteer programs, and in-depth interviews to ascertain whether older adults’ motivation to volunteer can improve the psychological well-being of older Korean immigrants and allow them to contribute in meaningful ways to community and society.

The number of older adults is predicted to increase rapidly in the near future. Future research should build on the current findings and further enhance recognition of the multidimensional determinants for volunteering among older Koreans in both countries. In addition, future studies should use a mixed methodology to qualitatively assess the rate of volunteering and the factors that contribute to volunteering among older Koreans in the United States and those in Korea. In addition, older Koreans in Korea are less likely to be engaged in volunteering, so additional research is needed to understand the factors that can explain this finding. Future study is needed to broaden the qualitative measurements and allow the findings of this study to be generalized. As it is, this study provides insight into what factors correlate with the rate of volunteering among older Koreans in both countries. As already mentioned, research to date has been sparse, and more detailed information is needed regarding the specific context of volunteering among older Koreans. This study deserves greater attention from researchers, gerontologists, policy analysists, and social workers given that they are now facing rapidly growing older populations in both countries.

## Figures and Tables

**Table 1 ijerph-19-07297-t001:** Socio-demographic characteristics of the sample (US: *n* = 240 & Korea: *n* = 240).

Variables	US(N)	(*n* = 240) %	Korea (N)	(*n* = 240) %
Gender				
Male	120	50	120	50
Female	120	50	120	50
Age				
60–69	120	50	120	50
75–79	120	50	120	50
Education				
Less High	65	27.1	117	48.8
High School	83	34.6	70	29.2
College	76	31.7	53	22.1
Graduate	16	6.7	7	2.9
Income				
Less $1000	153	65	160	66.7
$1000–$1999	46	19.2	41	17
$2000 or above	38	15.8	39	16.3
Marital Status				
Married	174	72.5	162	67.5
Sep/Div/Wid/N.m	66	27.5	78	32.5
Health Status				
Very poor or Poor	90	37.1	78	32.5
Fair	102	42.5	123	51.3
Good or very good	48	19	39	16.3
Religion				
No religion	56	23.3	43	17.9
Protestant	140	58.3	117	48.8
Catholic	33	13.8	35	14.6
Buddhist or Others	11	4.6	45	18.7
Perceived financial status				
Very bad or Bad	110	45.8	129	53.8
Fair	110	45.8	92	38.3
Good & very good	20	8.3	19	7.9
Income sources				
Public assistance	144	60	87	36.3
Employment	78	32.5	57	23.8
Assistance from child & Asset	18	7.5	96	40
Volunteering				
Yes	56	23.3	34	14.2
No	184	76.7	206	85.8

**Table 2 ijerph-19-07297-t002:** Contingency table of various independent variables with volunteering (US: *n* = 240 & Korea: *n* = 240).

Variables			US		(*n* = 240)	Korea		(*n* = 240)	
Volunteering			Yes (%)		No (%)	Yes (%)		No (%)	
Gender						Gender			
Male			27(22.5)		93(77.5)	20(16.7)		100(83.3)	
Female			29(24.2)		91(75.8)	14(11.7)		20(16.7)	
Age						Age ***			
60–64			14(28.0)		36(72.9)	19(27.1)		51(72.9)	
65–69			20(28.6)		50(71.4)	8(16.0)		42(84.0)	
70–74			16(21.9)		57(70.1)	7(10.5)		60(89.6)	
75–79			6(12.8)		41(87.2)	0(0.00)		53(100)	
Education ***						Education ***			
Less High			10(15.4)		88(84.6)	9(8.2)		101(91.8)	
High School			11(13.3)		72(86.8)	8(11.4)		62(88.6)	
College			27(35.6)		49(64.5)	13(24.5)		40(75.5)	
Graduate			8(50.0)		8(50.0)	4(57.1)		3(42.9)	
Income ***						Income ***			
Less $1000			4(4.7)		81(95.3)	2(2.4)		83(97.7)	
$1000–$1999			17(21.8)		61(78.2)	9(11.5)		69(88.5)	
$2000–$2999			22(40.7)		32(59.3)	11(25.6)		32(74.4)	
$3000 or above			13(56.5)		10(43.5)	12(35.3)		22(64.7)	
Marital Status ***						Marital Status *			
Married			51(29.3)		123(70.7)	30(18.5)		132(81.5)	
Sep/Div/Wid/N.m			5(7.6)		61(92.4)	4(5.1)		74(94.9)	
Health Status ***						Health Status ***			
Very poor			0(0.0)		14(100)	0(0.0)		10(100)	
Poor			7(9.2)		69(90.8)	3(4.4)		65(95.6)	
Fair			26(25.5)		76(74.5)	18(14.6)		105(85.4)	
Good or very good			23(47.9)		25(52.1)	13(33.3)		26(66.7)	
Perceived financial status ***						Perceived financial ***			
Very bad			0 (0.0)		14 (100)	0(0.0)		21(100)	
Bad			6 (6.3)		90 (93.8)	2(1.9)		106(98.2)	
Fair			37(33.6)		73 (66.4)	27(29.4)		65(70.7)	
Good & very good			13(65.0)		7 (35.0)	5(26.3)		14(73.7)	
Living alone ***						Living alone			
			2(3.8)		454(28.8)	5(9.1)		29(15.7)	
Living with spouse only ***						Living with spouse only			
			54(30.2)		2(3.3)	28(17.5)		6(7.7)	
Living with grandchild only						Living grandchild only			
			2(11.1)		54(23.3)	2(20.2)		32(13.9)	
Income resource						Income resource ***			
Employment			22(18.3)		98(81.7)	5(6.9)		68(93.2)	
Public assistance			27(26.5)		75(73.5)	15(21.7)		54(78.3)	
Assistance from child & Asset			7(25.1)		11(74.9)	14(32.3)		84(67.3)	
Religion *						Religion			
Yes			49(26.6)		135(73.4)	31(15.7)		166(84.3)	
No			7(12.5)		49(87.5)	3(7.0)		40(93.0)	
Elder care *						Elder care			
Disagree			42(31.3)		89(69.1)	6(11.3)		47(88.7)	
Agree			14(14.3)		84(85.7)	23(17.9)		105(82.0)	
Strongly agree			0(0.0)		11(100.0)	5(8.5)		54(91.5)	
Social network ***						Social networks			
0			14(12.5)		98(97.5)	4(8.9)		41(91.1)	
1–2			36(29.8)		85(69.2)	18(14.7)		105(85.3)	
3 or More			6(85.7)		1(14,3)	12(16.7)		60(83.3)	
Social support ***						Social support			
Seldom			33(16.8)		162(83.2)	2(9.5)		19(90.5)	
Often			21(51.2)		20(48.8)	26(15.5)		142(84.5)	
Always			2(50.0)		2(50.0)	2(7.4)		25(92.6)	
**US**	**Mean**	**SD**	**Min**	**Max**	**Korea Mean**	**SD**	**Min**	**Max**	***t*-test**
Social support	20.56	6.53	10	47	26.65	10.34	10	50	7.71 ***
Social networks	13.7	4.75	6	24	17.37	6.58	6	31	6.99 ***
Attitude of elder care	28.75	3.85	20	40	24.81	4.87	10	30	−9.83 ***

*** *p*< 0.001, * *p* < 0.05.

**Table 3 ijerph-19-07297-t003:** Types of volunteering and hours (monthly) (US: *n* = 240 & Korea: *n* = 240).

	US	*n*	%	Korea	*n*	%
Formal Volunteering						
	Yes	13	5.4	Yes	23	9.6
	No	226	94.2	No	217	90.4
Informal Volunteering						
	Yes	63	25.8	Yes	33	13.8
	No	178	74.1	No	207	86.2
Hours (monthly)						
	1	23	9.6	1	9	3.8
	2	28	11.7	2	15	6.3
	3 or more	16	6.7	3 or more	10	4.2

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
