# Peer review of "Comparative Study on Volunteering among Older Korean Immigrants in the United States and Older Koreans in South Korea"

_ijerph, 2022, doi:10.3390/ijerph19127297_

Round 1

Reviewer 1 Report

Let the paper be published.

Author Response

I very much appreciate the reviewers’ comments, which have been very helpful in improving the manuscript. 

1) In response to reviewer 1, I checked spell and revised the manuscript using the version of my manuscript found at the above link. However, I was not used “Track Changes” function.

Reviewer 2 Report

I have just two small comment, which can be corrected easily

row 329: t-test is not in table 2, please delete this but in row 326-327 in table 1 there is t test  but it is a comparison of two subgroups in the sample, so it is a bivariate method so I recommend to put into the bivariate analysis part I accept the revised paper, both theoretically and methodologically it is better now

Author Response

I very much appreciate the reviewers’ comments, which have been very helpful in improving the manuscript. 

1) In response to reviewer 2 comments, I checked spell and revised the manuscript using the version of my manuscript found at the above link. However, I was not used “Track Changes” function.

In response to reviewer 2 comments, I deleted t-test in Table 1(before row326-327) and moved to Table 2(row338).

This manuscript is a resubmission of an earlier submission. The following is a list of the peer review reports and author responses from that submission.

Round 1

Reviewer 1 Report

The study aimed to determine the most important factors that influence the prevalence of volunteering among older Koreans in the United States and in Korea and to identify sociodemographic resources, cultural factors, and social resources that are correlated with volunteering. The research sample is appropriate. Research method correctly applied. Properly conducted discussion and conclusions drawn.

My remarks refer mainly to the references since the research sample is appropriate, research method correctly applied, properly conducted discussion and conclusions drawn.
Here about the references:
- in items 1, 15, 18, 21 & 23, the title of the article (as opposed to the title of the journal) should be written in lowercase;
- in item 3, the volume numbering (i.e. 18) is missing, and the page numbering is incorrect (it should be 479–489 instead of 1-11);
- in items 8 & 17, there is no need to write the title of the journal in capital letters (it should be ‘Voluntas’ instead of ‘VOLUNTAS’);
- in item 9, page numbering should be 1213 instead of 1-30;
- in item 9, the words of the journal title should start with capital letters (i.e. ‘BMC Public Health’ instead of ‘BMC public health’);
- in item 15, the title of the journal is ‘Health and Welfare Policy Forum’ instead of ‘Korea Institute for Health and Social Affairs’, as well as the numbering of the volume should be ‘10 (300)’ instead of ‘10’;
- in item 16, the words of the journal title should start with capital letters (i.e. ‘American Sociological Review’ instead of ‘American sociological review’);
- in item 16, the volume number (i.e. 62) and the issue number (i.e. 5) are missing;
- in item 18, the journal numbering should be ‘3(Suppl 1)’ instead of ‘3’, and the page numbering should be simply ‘S163’;
- in item 22, journal numbering should be ‘32(4)‘ instead of ‘32’;
- in item 23, the editors of the book are not mentioned (i.e. Longtao He, Jagriti Gangopadhyay) in which the indicated article was contained; there is also a lack of a book publisher;
- in item 26, journal numbering should be ‘7(1)’ instead of ‘7’; page numbering should be ‘18870’;
- in item 27, journal numbering should be ‘132(3)’ instead of ‘132’; the doi number should also be added: 10.1080/00224545.1992.9924704;
- in item 28, the doi number should be added: 10.1093/geront/46.4.503;
- in item 29, journal numbering should be ‘13(2)’ instead of ‘2’;
- in item 30, the word ‘In’ should be removed before the title of the journal; the doi number should also be added: https://doi.org/10.1111/socf.12208.
The notation of the year of the journal should also be standardized – once in the bibliography it is placed after the journal title, and at other times it is placed in parentheses and inserted after the names of the authors. In general, the record of sources should be adapted to the editorial guidelines of the journal.

Reviewer 2 Report

This should be important research in developing a successful aging society.

In order to make it clearer, please consider these points.
1) Please describe the detail of the questionnaire wording about the participation in volunteering.
2) I did not understand the meaning of internal consistency of "volunteering measurement" (with Cronbach’s alpha).
3) Please explain the reason why you adopted the variable of "Attitude of eldercare" in this analysis.
4) Please insert the explanation about inclusion criteria of age (>60 YO) nearby objective.
5) I wonder whether the prevalence of age (60-69:70-79  = 50:50) followed your strategy or not. Because it looks so unnatural, then please add explain (shortly OK).
6) I recommend you add a detailed context of "volunteering" in both countries. For example, accessibility, prevalence, opportunity to participate, and definition of volunteering. I know your RQ is focused on individual factors and related concepts, but for readers, I think the information about the research setting environment is important to understand your result. 

Reviewer 3 Report

I do not recommend the publication of this paper as it is below standard both theoretically and methodologically.

Ideally, the study should have an introduction (setting out the topic and its relevance, the research questions, and the structure of the paper), with the theoretical background provided separately. The theoretical section should discuss definitions, introduce the difference between formal and informal volunteering, explore recent trends in volunteering, and finally, present the theories on micro-level determinants of volunteering. Discussing Wilson’s resource theory is not sufficient (partly because the effects of religion and values should be examined as well).

The inclusion of the health status variable and the Attitude of Elder Care scale in the empirical part is questionable as it is not supported by a related theory, research question, and hypothesis.

Since convenience sampling was used as opposed to probability sampling, the sample is not representative, so multivariate methods (such as logistic regression) are not recommended; instead, the empirical analysis should be restricted to bivariate methods. The self-reported measurement of volunteering does not have a clear definition, so we cannot be certain whether it includes informal volunteering in addition to formal volunteering. Cronbach’s alpha indicator is not needed with respect to volunteering. The presentation of variables is incomplete and occurs in multiple instances in tabular form only. Even though only continuous and standardised variables can be correlated, the correlation analysis of the paper features several nominal variables, which is a mistake. For these, cross tabulation would be sufficient.

In the summary section, the results of bivariate analyses should be separated from regression results. This is because, given the correct sampling method, regression results are to be preferred over bivariate analyses.